# A Novel Solid Nanocrystals Self-Stabilized Pickering Emulsion Prepared by Spray-Drying with Hydroxypropyl-β-cyclodextrin as Carriers

**DOI:** 10.3390/molecules26061809

**Published:** 2021-03-23

**Authors:** Jifen Zhang, Yanhua Wang, Jirui Wang, Tao Yi

**Affiliations:** 1College of Pharmaceutical Sciences, Southwest University, Chongqing 400716, China; wangyanhua@email.swu.edu.cn; 2Chongqing Academy of Chinese Materia Medica, Chongqing 400065, China; wangjiruizyy@163.com; 3Macao Polytechnic Institute, School of Health Sciences and Sports, Macao 00853, China

**Keywords:** nanocrystals, Pickering emulsions, solid dosage form, hydroxypropyl-β-cyclodextrin, spray-drying, puerarin, poorly soluble drugs

## Abstract

A drug nanocrystals self-stabilized Pickering emulsion (NSSPE) with a unique composition and microstructure has been proven to significantly increase the bioavailability of poorly soluble drugs. This study aimed to develop a new solid NSSPE of puerarin preserving the original microstructure of NSSPE by spray-drying. A series of water-soluble solid carriers were compared and then Box-Behnken design was used to optimize the parameters of spray-drying. The drug release and stability of the optimized solid NSSPE in vitro were also investigated. The results showed that hydroxypropyl-β-cyclodextrin (HP-β-CD), rather than solid carriers commonly used in solidification of traditional Pickering emulsions, was suitable for the solid NSSPE to retain the original appearance and size of emulsion droplets after reconstitution. The amount of HP-β-CD had more influences on the solid NSSPE than the feed rate and the inlet air temperature. Fluorescence microscopy, confocal laser scanning microscopy and scanning electron microscopy showed that the reconstituted emulsion of the solid NSSPE prepared with HP-β-CD had the same core-shell structure with a core of oil and a shell of puerarin nanocrystals as the liquid NSSPE. The particle size of puerarin nanocrystal sand interfacial adsorption rate also did not change significantly. The cumulative amount of released puerarin from the solid NSSPE had no significant difference compared with the liquid NSSPE, which were both significantly higher than that of puerarin crude material. The solid NSSPE was stable for 3 months under the accelerated condition of 75% relative humidity and 40 °C. Thus, it is possible todevelop the solid NSSPE preserving the unique microstructure and the superior properties in vitro of the liquid NSSPE for poorly soluble drugs.

## 1. Introduction

The low bioavailability is a great challenge for oral administration of poorly water-soluble drugs, such as puerarin, silybin and so on. A novel liquid Pickering emulsion with a unique composition and microstructure, i.e., drug nanocrystals self-stabilized Pickering emulsion (NSSPE) has been developed to orally deliver poorly water-soluble drugs. In NSSPE, nanocrystals of poorly water-soluble drugs, which acted as the therapeutic substance as well as the stabilizer of emulsion, were adsorbed at the oil-water interface, forming a unique core-shell structure with a core of oil saturated with drug and a shell of drug nanocrystals [1,2]. Consequently, NSSPE could significantly enhance the oral bioavailability of poorly water-soluble drugs. Pharmacokinetics studies showed NSSPE improved AUC of silybin and puerarin to 4.6 times and 2.6 times compared with crude material, respectively [1,2], suggesting great application potential for oral administration of poorly water-soluble drugs. NSSPE also has the higher drug loading and safety than traditional emulsions [3,4].

Compared with liquid preparations, solid preparations are more stable and more suitable for industrial large-scale production, transportation, storage and clinical application. There have been a few reports of solid Pickering emulsions with heterogeneous solid particles as stabilizers [5,6,7]. In view of the unique composition and microstructure of the new NSSPE, it was a greater challenge to develop solid NSSPE with the same advantages as liquid NSSPE for oral poorly soluble drugs.

Spray drying is a common method for preparing dry emulsions [8,9]. Starch [10], lactose [11], maltodextrin [12], mannitol [13], dextran [14] and hydroxypropyl methylcellulose [15] are commonly solid carriers used in spray drying of emulsions. The solid carrier may have important impact on the properties of spray-dried emulsions. Silica nanoparticles with a particle size of 10 nm could be used as a solid carrier to produce spray-dried Pickering emulsions with good reconstitution [16]. However, using the similar silica nanoparticles with a particle size of 30 nm, the spray-dried Pickering emulsion showed irregular fractal-like aggregates, with each aggregate consisting of several oil droplets compressed together and could not form the same reconstituted emulsion as the original emulsion when dispersed in water [5]. Moreover, the same solid carrier may cause spray-dried emulsions with different reconstitution properties for Pickering emulsions of different compositions. Using the same maltodextrin (12.0%, *w*/*w*) as the solid carrier, the properties of the spray-dried emulsions obtained from two similar Pickering emulsions with different protein stabilizers were quite different [17]. The droplet size distribution of reconstituted emulsions was affected significantly by the concentration and type of protein stabilizers.

The NSSPE of puerarin composed of puerarin nanocrystals as solid particle stabilizers and *Ligusticum chuanxiong* essential oil as the main oil phase [2], was significantly different from traditional Pickering emulsions. Firstly, puerarin nanocrystals adsorbed at the oil-water interface were significantly different from other solid particle stabilizers in structure (Figure 1A) and property. Secondly, *Ligusticum chuanxiong* essential oil was completely different from other chemical fatty oils, which was mainly composed of small molecules, such as ligustilide (Figure 1B), senkyunolide A (Figure 1C) and less than 10% of fatty acid and esters [2]. As a result, it was a worthy research whether solid NSSPE prepared with commonly used solid carriers could restore structures and properties of the original liquid NSSPE. Since the advantages of NSSPE in vitro and in vivo, including improvement of stability and oral bioavailability, were largely resulted from the unique core-shell microstructure [1,2], choosing a suitable solid carrier to protect the unique microstructure of NSSPE was a key issue in the preparation of the solid NSSPE by spray drying.

The purpose of this paper was to develop a new solid NSSPE of puerarin by spray drying. An appropriate solid carrier was selected from a series of carriers to retain the special microstructure and properties of liquid NSSPE. Box–Behnken design was used to explore the effects of spray-drying process on reconstitution properties of the solid NSSPE. Dissolution in vitro and physical stability of the solid NSSPE were also investigated.

## 2. Materials and Methods

### 2.1. Materials

Puerarin of 98% purity was purchased from Sichuan Yuxin Pharmaceutical Co., Ltd. (Chengdu, China). Labrafil M 1944 CS was purchased from GATTEFOSSé (Shanghai, China) Trading Co., Ltd. (Shanghai, China). *Ligusticum chuanxiong* oil (with a content of ligustilide 47.12%, senkyunolide A 22.51% and other components all less than 5%) was purchased from Jiangxi Xuesong Natural Medicinaliol Co., Ltd. (Ji’an, China). Hydroxypropyl-β-cyclodextrin (HP-β-CD, molecular weight of 1541.54) was purchased from Shanghai Yuanye Biotechnology Co., Ltd. (Shanghai, China). α-cyclodextrin (α-CD, purity of 98%) was purchased from Shanghai Aladdin Biochemical Technology Co., Ltd. (Shanghai, China). CAVAMAX- W6 (γ-CD, pharma grade) was purchased from Wacker chemical (China) Co., Ltd. (Shanghai, China). β-cyclodextrin (β-CD), starch dextrin, mannitol and lactose were purchased from Chengdu Kelong Chemical Reagent Factory (Chengdu, China). Maltodextrin was purchased from Chongqing Yuexiang Chemical Co., Ltd. (Chongqing, China). Nile red (purity >98%) was purchased from Sigma (St. Louis, MO, USA) and Nile blue (purity >75%) was purchased Adamas Co., Ltd. (Shanghai, China). Methanol and acetonitrile of HPLC-grade were purchased from Sigma (St. Louis, MO, USA). Other solvents and chemicals were of an analytical grade.

### 2.2. Preparation of Liquid NSSPE

Liquid NSSPE of puerarin was prepared according to our previous studies [9]. Briefly, 350 mg of puerarin and 63 mL of pure water whose pH was pre-adjusted to 11.5 with 1 mol·L^−1^NaOH were mixed by a high-speed shearing machine (FA25, FLUKO fluid machinery manufacturing Co., Ltd., Shanghai, China) at 13000 rpm for 2 min, followed by a high pressure homogenizer (AH100D, ATS Engineering Ltd., suzhou, China) at 80 MPa for 15 cycles to prepare puerarin nanosuspension. Then, 7 mL of mixed oil of *Ligusticum chuanxiong* essential oil and Labrafil M 1944 CS (9:1, *v*/*v*) was added and continued to be homogenized at 80 MPa for another 15 cycles.

### 2.3. Selection of the Carrier for Solid NSSPE

#### 2.3.1. Preparation of Solid NSSPE

Water soluble solid carriers, including starch dextrin, maltodextrin, mannitol, lactose as well as α-cyclodextrin, β-cyclodextrin, HP-β-CD and γ-cyclodextrin (10%, *w*/*v*) were added into 20 mL of liquid NSSPE, respectively. After the solid carrier dissolved completely, the resultant mixture was spray dried using a spray drier (B-290, BuchiLabortechnik AG, Flawil, Switzerland). The feed rate was 3.5 mL·min^−1^ with pump parameter of 15, the air flow was 600 L·h^−1^ and the inlet air temperature was 120 °C.

#### 2.3.2. Reconstitution Properties of Solid NSSPE

The solid NSSPE of 0.5 g was dispersed with 5 mL of pure water by shaking slightly for 30 s. The appearance of reconstituted liquid emulsion was observed to evaluate reconstitution property. The droplet size of reconstituted emulsions was also measured directly with a particle size analyzer (BT-9100, better size instruments Ltd., Dandong, China). The refractive index of solid particles was set to 1.52. The refractive index of the continuous phase was set to 1.33 which was the refractive index of water. The obscuration was between 10% and 20%.

#### 2.3.3. Microstructure Analysis of Solid NSSPE with HP-β-CD

To test whether HP-β-CD was a suitable solid carrier, the microstructures of solid NSSPE prepared with HP-β-CD and its reconstituted emulsion were studied further.

##### Morphological Analysis of Solid NSSPE

The outer microscopic structure of the solid NSSPE was observed by SEM with a scanning electron microscope (JSM-6510LV, JEOL Ltd., Tokyo, Japan). The sample was gently blown on a silicon wafer and coated with a thin layer of gold.

##### Fluorescence Microscopy and Confocal Laser Scanning Microscopy

Both the liquid NSSPE and the reconstituted emulsion from the solid NSSPE were observed by fluorescence microscopy (FM) and confocal laser scanning microscopy (CLSM). One drop of emulsion sample was spotted on a glass slide and observed directly with a fluorescence microscope (DFC310 FX, Leica, Solms, Germany). For CLSM observations, both emulsions were labeled with a fluorescent dye mixture of Nile red alcohol solution and Nile blue A aqueous solution [18,19]. The labeled emulsion was observed at excitation wavelengths of 488 nm for Nile Red and 633 nm for Nile Blue A with a Nikon N-STORM CLSM (Nikon, Tokyo, Japan).

##### Characterization of Puerarin Nanocrystals Adsorbed at the Oil–Water Interface

Puerarin nanocrystals at the interface of emulsion droplets were separated from emulsions according to a literature method with slight modification [20,21]. A certain amount of the liquid NSSPE and reconstituted emulsion from the solid NSSPE were centrifuged at 165,654× *g* for 1 h at 4 °C to separate oil phase (top), aqueous phase (bottom) and adsorbed nanocrystals (middle).

The aqueous phase and oil phase were taken out very carefully. The volume and drug content in two phases were measured and the adsorption percentage of drug nanocrystals was calculated according to the following formula:Adsorption percentage of drug nanocrystals (%) = (M_e_ − M_o_ − M_w_)/M_e_(1)

M_e_ represented the drug amount in emulsion, M_o_ and M_w_ represented the drug amount in oil layer and aqueous phase after centrifugation, respectively.

At the same time, the resultant substance in the middle layer was put in an eppendorf tube and dispersed in 1 mL of water by vortexing and ultrasonication. Then, the resultant sample was analyzed by a laser particle size analyzer (Zetasizer Nano-ZS, Malvern Instruments, Malvern, UK).

To validate the crystallinity of puerarin after formulation with a pH of 11.5, the resultant substance in the middle layer was collected and dried at room temperature. As a control, Puerarin nanocrystals were collected by filtering the puerarin nanosuspension through 100 nm polycarbonate membrane and dried at room temperature. Differential scanning calorimetry (DSC)analysis was performed using a DSC 200PC (Netzsch Ltd., Selb, Germany). With approximately 5 mg of samples sealed in an aluminum pan, the thermal behavior was analyzed from 60 °C to 240 °C using a heating rate of 10 °C/min.

### 2.4. Optimization of Spray Drying Process by Box–Behnken Design

Preliminary experiment showed that the amount of solid carrier (*X_1_*), the feed rate (indicated by the feed pump parameter, *X_2_*) and the inlet air temperature (*X_3_*) had effects on the reconstitution properties of the solid NSSPE. Therefore, these three factors were selected to be independent variables for further optimization. The oil leakage rate of reconstituted emulsion (*Y_1_*), droplet size of reconstituted emulsion(*Y_2_*) and drug content in the solid NSSPE (*Y_3_*) were chosen as dependent variables.

#### 2.4.1. Oil Leakage Rate of Reconstituted Emulsions

When different solid NSSPEs were re-dispersed in water, some formed uniform emulsions and others showed obvious oil leakage in the top (Figure 1 and Figure 2). To separate the leaked oil, reconstituted emulsion was centrifuged at 1800× *g* for 15 min. Then, the leaked oil was collected carefully with a micro syringe. The collected oil was dissolved to 5 mL with ether. A total of 100 μL of the ether solution was mixed with 400 μL of methanol for analysis. The content of ligustilide was determined by HPLC method which had been verified according to the Chinese Pharmacopoeia of 2015 edition. An Agilent ZORBAX SB-C_18_ column (4.6 mm × 250 mm, 5 µm) was used. The mobile phase was composed of 0.1% formic acid in water (A) and methanol (B) with a gradient elution as follows: 0–5 min, 45–60% of B; 5–15 min, 60–70% of B; 15–25min, 70–75% of B and 25–30 min, 75–45% of B. The flow rate was 1 mL·min^−1^ and the detection wavelength was 254 nm.
(2)$$\mathrm{Oil}\;\mathrm{leakage}\;\mathrm{rate}=\frac{\mathrm{Amount}\;\mathrm{of}\;\mathrm{ligustilide}\;\mathrm{in}\;\mathrm{the}\;\mathrm{oil}\;\mathrm{leakage}\;\mathrm{of}\;\mathrm{recomstituted}\;\mathrm{emulsion}}{\mathrm{Percent}\;\mathrm{of}\;\mathrm{ligustilide}\;\mathrm{in}\;\mathrm{the}\;\mathrm{mixed}\;\mathrm{oil}\;\times \mathrm{the}\;\mathrm{mass}\;\mathrm{of}\;\mathrm{mixed}\;\mathrm{oil}\;\mathrm{in}\;\mathrm{the}\;\mathrm{NSSPE}}\times 100\%$$

#### 2.4.2. Drug Content in the Solid NSSPE

The solid NSSPE of 0.2 g was dispersed in 2 mL of pure water firstly. Then, 100 μL of the reconstituted emulsion was withdrawn into 500 μL of methanol-chloroform (1:2, *v*/*v*) and sonicated for 5 min. The finial volume was adjusted to 10 mL with methanol. The puerarin concentration was measured by HPLC (LC-20A HPLC system, Shimadzu Corporation, Kyoto, Japan) with a SilGreen^®^ GH0525046 C_18_ column (4.6 mm × 250 mm, 5 µm) after filtered through a 0.22 μmmicroporous filter membrane. The mobile phase was water-acetonitrile (85:15, *v*/*v*) and flow rate was 1 mL·min^−1^. The detection wavelength was 250 nm [2].

### 2.5. In Vitro Dissolution Test

The dissolution test was performed by dialysis method using a USP II apparatus at 37 °C with a paddle speed of 50 rpm. Puerarin crude material and the solid NSSPE with the same puerarin content as the liquid NSSPE were put in dialysis bags (sigma) with a molecular cut off of 8–14 kDa, respectively and then 2 mL of pure water was added. 2 mL of the liquid NSSPE was directly placed into a dialysis bag. All dialysis bags were sealed and placed in 100 mL of medium which was 0.1 M hydrochloric acid solution for the first 2 h and adjusted pH to 6.8 with 1M NaOH after 2 h. Samples were collected at appropriate time intervals and filtered through a 0.22 μm filter. The concentration of puerarin in each sample was determined by HPLC and drug release percentage was calculated.

### 2.6. Physical Stability of Solid NSSPE

The solid NSSPE was sealed and stored under an accelerated condition of 40 °C and 75% of relative humidity for 3 months. The appearances of the solid NSSPE powder and its reconstituted emulsion were observed at 0, 1, 2 and 3 months, respectively. Droplet size of reconstituted emulsion and drug content were also determined.

### 2.7. Statistical Analysis

All data were expressed as mean ± standard deviation (SD). One-way analysis of variance was used to test differences between groups and *p* < 0.05 or *p* < 0.01 were considered to be significant difference.

## 3. Results and Discussions

### 3.1. Effects of Solid Carriers on Reconstitution of Solid NSSPE

Commonly used solid carriers, including starch dextrin, maltodextrin, mannitol, lactose, were firstly used to prepare solid NSSPE. The appearance and droplet size of reconstituted emulsions from these solid NSSPE were shown in Figure 2.

It was found that solid NSSPE prepared with the most commonly used solid carriers in spray drying of emulsions, such as starch dextrin, maltodextrin, mannitol and lactose, all showed poor reconstitution. Significant leakages of oil phase were observed for these four reconstituted emulsions as shown in Figure 2B. The droplet sizes of reconstituted emulsions from these solid NSSPE increased significantly and were 4.82, 7.84, 2.32 and 6.80 times that of the liquid NSSPE, respectively. This may be because that these solid carriers could not protect the microstructure of the liquid NSSPE during spray drying. It was difficult for the oil in these solid NSSPE powders to be dispersed uniformly without sufficient energy provided by high pressure homogenization when re-dispersed in water. Consequently, significant increase in droplet size of reconstituted emulsions was unavoidable (Figure 2C).

In recent years, Pickering emulsions stabilized by cyclodextrin have been reported [22,23,24] for drug delivery, suggesting that cyclodextrin was beneficial to the construction and stability of Pickering emulsion. HP-β-CD [25], γ-CD [26] and β-CD [27] also have been used to solid emulsions, omega-3 and olive oil, respectively. Consequently, solid NSSPE was further prepared with four cyclodextrins as solid carriers, respectively. As shown in Figure 3, contrary to commonly used solid carriers mentioned above, no obvious oil was observed in all reconstituted emulsions from solid NSSPE with cyclodextrins as solid carriers. This may be due to the inclusion effect of cyclodextrin. The oil could be included into the surrounding cyclodextrin even if it was leaked from emulsion droplets. However, there were still many differences for the four cyclodextrins. As far as α- and β-CD were concerned, the reconstituted samples nearly had no emulsion characteristics, with obvious sediment in the bottom, clear water in the middle and a lillte cyclodextrin on the top. For γ-CD, the emulsion characteristics still remained to a certain extent, but sediment also appeared. Only the solid NSSPE with HP-β-CD could reconstitute a uniform liquid emulsion with a droplet size similar to that of the liquid NSSPE. These results suggested that structure of cyclodextrins may have influence on the reconstitution property of the solid NSSPE. The potential mechanism may be related to the intramolecular cavity and the solubility of cyclodextrin.

As reported, when cyclodextrins were used as solid particles to stabilize Pickering emulsions, the inclusion stability of oil molecules in cyclodextrin cavity is a determining parameter in thermal stability of Pickering emulsions [28]. Hu J.W. studied the influences of the relationship between cyclodextrins and oils on fabrication of Pickering emulsions. Medium chain triglyceride (MCT) and castor oil both belong to triglyceride oils, but MCT could form a stable structure of the Pickering emulsion only withα-CD, while castor oil could form a stable Pickering emulsion with both α-CD and β-CD. For ring-structured oils, including limonene and octisalate, only octisalate could form a stable Pickering emulsion, while separation occurred after one minute with limonene for either α-CD orβ-CD. These results demonstrated that stable Pickering emulsions could be formed only when the oil structure was adapted to intramolecular cavity of the cyclodextrin and hydrogen bond was formed between the cyclodextrin and oil [29]. α-, β- and γ-CD are composed of 6, 7and 8 α-d-glucopyranose units, respectively. HP-β-CD is a derivative of β-CD with the hydrogen atom in the 2-, 3- or 6-hydroxyl group of glucose replaced by hydroxypropyl group as shown in Figure 1D. As a result, the cavity size of HP-β-CD is different from α- and γ-CD and hydrophilicity of cavity surface of HP-β-CD is stronger than that of β-CD. The intramolecular space of HP-β-CD may be more suitable for *Ligusticum chuanxiong* essential oil used here. The hydroxyl groups in hydroxypropyl group of HP-β-CD may also form hydrogen bond with the carbonyl group in ligustilide and senkyunolide A which account for about 70% of the mixed oil phase, further stabilizing the inclusion compound.

Only reconstituted emulsion of HP-β-CD had no sediment. This may be due to the highest solubility of HP-β-CD, which was at least 3.4, 27.0 and2.2 times that of α-, β- and γ-CD, respectively [30].When the solid NSSPE was dispersed in water, HP-β-CD was easy to dissolve in water and then oil in the inclusion of HP-β-CD could be dispersed more uniformly. Simultaneously with the aid of drug nanocrystals adsorbed at the oil-water interface, which was confirmed in subsequent studies on the microstructure of the reconstituted emulsion, the emulsion with similar appearance and droplet size to the liquid NSSPE could be reconstituted even if without high energy provided by high pressure homogenization. To the best of our knowledge, no other Pickering emulsions or emulsions containing volatile oil from Traditional Chinese Medicine were spray dried with cyclodextrins as solid carriers. A clearer mechanism still requires further in-depth study.

### 3.2. Microstructure Analysis of the Solid NSSPE with HP-β-CD

The SEM images of the solid NSSPE powder prepared with the optimized spray-drying parameters were shown in Figure 4. The solid NSSPE powder had a good spherical shape with particle size of about 10–30 μm. The particle surface was not smooth with obvious wrinkles and potholes on it. When water in Pickering emulsion was evaporated rapidly during spray drying, the particles adsorbed at the oil-water interface could form a shell-like structure surrounding the oil droplets and the solid carrier could form a bridge structure to stabilize the solid Pickering emulsion [16]. Whitby’s research [7] also showed a similar result to ours. They found that the dried emulsion powder was shrinking spheres and the mean particle size of powder was larger than that of emulsion droplets. After HP-β-CD was added to liquid NSSPE, the oil droplets with adsorbedpuerarin nanocrystals were surrounded by HP-β-CD. On the other hand, dissolved puerarin in aqueous phase of NSSPE may be incorporated into HP-β-CD. During spray drying, the continuous phase of water was evaporated instantly, leaving the oil droplets surrounded by puerarin nanocrystals and HP-β-CD, both of which could stabilize reconstituted emulsion droplets together.

Reconstitution property of the solid NSSPE was one of the most important indexes for evaluating the solid NSSPE. For a good dried emulsion, the droplet size of reconstituted emulsion should remain the same as the original emulsion, which required that the emulsion droplets should not be destroyed during the drying process and be well preserved in the powder matrix [31]. The FM and CLSM observation (Figure 5 and Figure 6) showed that the reconstituted emulsion droplets had a good circle or oval shape. The droplet surface of reconstituted emulsion was surrounded by strong green fluorescence of puerarin, the same as the liquid NSSPE (Figure 5). The same fluorescence distributions were also observed by CLSM in the reconstituted emulsion from the solid NSSPE as the liquid NSSPE (Figure 6). The oil droplets labeled with Nile red appeared red and green fluorescence of Nile blue which was used to label puerarin nanocrystals was observed at the interface of the oil droplets. It could be concluded that even though HP-β-CD contributed to the stability of reconstituted emulsion, puerarin nanocrystals were still adsorbed at the oil-water interface to stabilize reconstituted emulsion, i.e., the microstructure of Pickering emulsion did not change significantly.

In addition, Table 1 verified not only the emulsion droplet size, but also the particle size and the adsorption rate of puerarin nanocrystals did not change significantly for the reconstituted emulsion from the solid NSSPE of puerarin.

DSC analysis was used to verify the crystallinity of puerarin. As shown in Figure 7, there was a sharp endothermic peak at 202 °C for puerarin crude material (Figure 7a), which was also observed for puerarin nanocrystals(Figure 7b) and adsorbed puerarin in liquid NSSPE (Figure 7c) at the similar temperature.It was suggested that the crystallinity of puerarin may not change for both puerarin nanocrystals and the liquid NSSPE. However, no obvious endothermic peak appeared in the DSC thermogram of reconstituted emulsion from the solid NSSPE. It may be due to the interference of HP-β-CD.As analyzed before, puerarin nanocrystals and HP-β-CD could stabilize reconstituted emulsion droplets together. After ultracentrifugation of reconstituted emulsion, the resultant substances in the middle layer may contain both puerarin nanocrystals and some HP-β-CD.

### 3.3. Optimization of Spray-Drying Process by Box-Behnken Design

In addition to properties of the emulsion including ratio of oil to water, emulsion viscosity and droplet size, spray-drying process, such as the type and amount of solid carrier, liquid feed rate and inlet air temperature may also affect reconstitution properties of the solid NSSPE [7,32,33]. In order to obtain solid NSSPE with good reconstitution property, a three-factor, three-level Box–Behnken design was employed to statistically optimize the spray-drying process. According to the design, 17 solid NSSPE were prepared with HP-β-CD and their reconstitution properties were correspondingly investigated. The appearances of 17 solid NSSPE and their reconstituted emulsions were shown in Figure 8. All the responses measured for 17 experimental runs were present in Table 2.

Figure 8 showed that solid NSSPE numbered 7, 10, 13 and 17were light yellow and obvious leakage of oil phase in their reconstituted emulsions appeared. Correspondingly, the oil leakage rates of these four samples were all over 1% and the droplet size of their reconstituted emulsions increased significantly. The yield rate of spray-drying was not used as a response to optimize spray drying because it may be influenced by the collecting operation. In fact, the yield rates of the 17 samples were in the range of 80% to 85%, without significant difference.

The regression analysis of the data in Table 2 was calculated by SPSS 11.0 software based on the nonlinear quadratic model. ANOVA analysis was used to identify significant parameters that affected responses and fitness of models. Model reduction by removing any nonsignificant model terms that were not needed to support hierarchy was adopted to improve the chosen model. The responses were analyzed individually and the following predicted equations were generated.
*Y_1_* = 4.101 − 0.715*X*_1_ − 0.0187*X*_2_ + 0.005*X*_3_ + 0.0298*X*_1_^2^ + 0.0015*X*_1_*X*_2_ − 0.0004*X*_1_*X*_3_ (*r* = 0.998, *F* = 253.224, *P* = 0.000)(3)
*Y_2_* = 27.398 − 1.644*X*_1_ − 0.19*X*3 + 0.05.821*X*_1_^2^ − 0.000459*X*_2_^2^ + 0.0006946*X*_3_^2^ (*r* = 0.994, *F* = 120.901, *P* = 0.000)(4)
*Y_3_* = 42.353 − 8.204*X*_1_ + 0.797*X*_3_ + 0.19*X*_1_^2^ − 0.00362*X*_3_^2^ + 0.0136*X*_1_*X*_3_ (*r* = 0.995, *F* = 133.890, *P* = 0.000).(5)

Three-dimensional response surface plots and contour maps of responses in function of two variables were generated by Origin 6.0 software based on the predicted equations, which were shown in Figure 9, Figure 10 and Figure 11.

Figure 9 showed that the main factor influencing the oil leakage rate of reconstituted emulsion was the amount of solid carrier. The oil leakage rate decreased rapidly and approximately linearly with the increase of the amount of solid carrier (*X*_1_) within the range of 5% to 10% when feed rate (*X*_2_) and inlet air temperature (*X*_3_) were set at a constant (Figure 9A,B). When the amount of solid carrier was above 10%, nearly no oil leaked. By contrast, increasing feed rate or decreasing inlet air temperature only could slightly reduce the oil leakage rate (Figure 9C) and the decline of oil leakage rate was nearly negligible.

Figure 10 showed that droplet size of reconstituted emulsion decreased rapidly with the increase of the amount of solid carrier within the range of 5% to 10%. However, when the amount of solid carrier exceeded 10%, the decrease of droplet size slowed down (Figure 10A,B). With the increase of the inlet air temperature, the droplet size of reconstituted emulsion showed a V-shaped change and reached the lowest value at 135 °C of inlet air temperature (Figure 10C). The feed rate had the smallest effect. The droplet size of reconstituted emulsion only decreased slightly with the increase of the feed rate (Figure 10A,C).

The drug content in solid NSSPE decreased greatly with the increasing of the amount of solid carrier (Figure 11A,B). The highest drug content in solid NSSPE of over 50 mg·g^-1^ could be reached when the amount of solid carrier was within 5–6.5%. Effect of the inlet air temperature on the drug content was accordance with a bell shape with the highest drug content at 130 °C of the inlet air temperature (Figure 11C). By contrast, the feed rate had little effect on the drug content (Figure 11A,C).

Altogether, within the studied range, the inlet air temperature and the feed rate only had very slight influences on the oil leakage rate, droplet size and drug content. On the contrary, the amount of solid carrier had much greater influence on the reconstitution properties of spray-dried NSSPE than the inlet air temperature and the feed rate. Both the oil leakage rate and droplet size in reconstituted emulsion declined rapidly with increase of the amount of solid carrier. These may be due to the protective effect of HP-β-CD on the oil droplet, as mentioned above. HP-β-CD as the solid carrier could prevent oil droplets from coalescence and oil leakage during spray-drying or storage.

According to three-dimensional surface plots and contour maps, the optimal values of independent variables were determined as follows: *X_1_* = 12%, *X_2_* = 23, *X_3_* = 130 °C. A model validation was performed by preparing three replicate batches on three different days based on the above optimal values. The yield rate was 84.3 ± 1.73%. As shown in Table 3, the actual values of three responses were close to or better than predicted values, indicating that optimization process was successful with a reliable predictive model.

### 3.4. In Vitro Dissolution Test

The in vitro dissolution profiles of puerarin crude material, the liquid NSSPE and solid NSSPE were shown in Figure 12. The drug release from both NSSPE was significantly faster than from puerarin crude material. During the first 120 min in stimulated gastric fluid, the dissolution rate of puerarin crude material was less than 30%, while the dissolution rates of the liquid NSSPE and solid NSSPE were 67.94 ± 3.16% and 54.78 ± 1.89%, respectively. After the following 180 min in stimulated intestinal fluid, 85.08 ± 2.45% and 82.49 ± 1.42% of puerarin was released from the liquid NSSPE and solid NSSPE, respectively, which were much higher than 58.88 ± 0.49% of puerarin crude material. These may be due to the special microstructure of puerarin nanocrystals adsorbing at the surface of oil droplets of NSSPE. Comparing the two NSSPE, the drug release from the liquid NSSPE was slightly faster than from the solid NSSPE. It may be due to the emulsion droplets being adsorbed by and wrapped in solid carrier for the solid NSSPE. The solid carrier must be dissolved first and then be followed by the release of drug. Therefore, the drug release from the solid NSSPE slowed down. However, the final cumulative release rate of the solid NSSPE was not significantly different from that of the liquid NSSPE. This was because the reconstituted emulsion from the solid NSSPE maintained the same microstructure as the liquid NSSPE. The emulsion droplet size, particle size and adsorption rate of puerarin nanocrystals of reconstituted emulsion were all consistent with those of the liquid NSSPE, seen from Table 1.

### 3.5. Physical Stability of Solid NSSPE

The solid NSSPE was sealed in vials and stored for 3 months under an accelerated condition of 40 °C, 75% relative humidity. Moisture absorption, discoloration, oil leakage and other unstable phenomena were not observed. The solid NSSPE could be reconstituted to a homogeneous emulsion without oil leakage after dispersed in water as shown in Figure 13A. Figure 13B verified that there were no significant changes in the droplet size and the drug content in the reconstituted emulsion. These results suggested that the solid NSSPE had a good stability for 3 months under the accelerated condition.

## 4. Conclusion

The present investigation has shown that it is possible to develop a new solid NSSPE of puerarin preserving the unique microstructure and the superior properties in vitro of liquid NSSPE byspray drying. The kind of solid carrier was found to be the key factor for successful solidification. Among the commonly used solid carriers and a series of cyclodextrins, HP-β-CD was found to be beneficial for solid NSSPE powder to reconstitute to a uniform emulsion with the same size and microstructure of droplets as the liquid NSSPE, as well as the adsorption rate of puerarin nanocrystals. This may be due to the suitable cavity structure, hydroxypropyl groups in the outer surface and higher solubility of HP-β-CD. The amount of HP-β-CD was another important factor. Finally, the optimal solid NSSPE could be obtained with 12% of HP-β-CD as solid carrier at a feed rate of 5.5 mL min^−1^ and inlet air temperature of 130 °C. The solid NSSPE had a similar cumulative drug release compared with the liquid NSSPE and could be stable for 3 months under an accelerated condition. In the future, further studies are necessary to evaluate the oral bioavailability of the solid NSSPE for poorly water-soluble drugs.

## Figures and Tables

**Figure 1 molecules-26-01809-f001:**
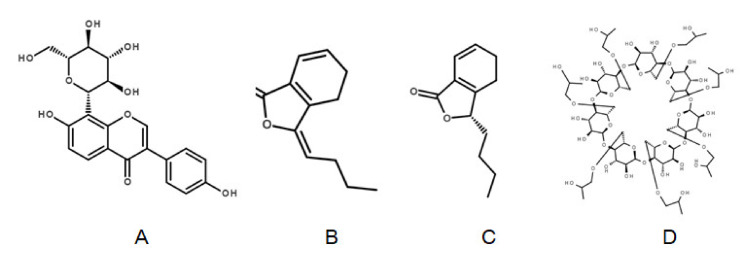
The structures of puerarin (**A**), ligustilide (**B**), senkyunolideA (**C**) andHydroxypropyl-β-cyclodextrin (**D**).

**Figure 2 molecules-26-01809-f002:**
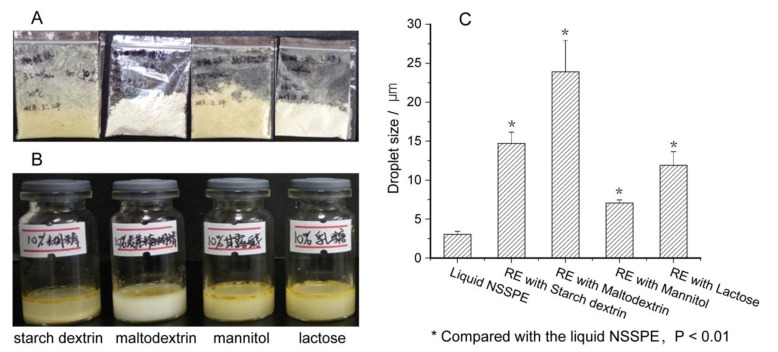
(**A**) Appearance of solid drug nanocrystals self-stabilized Pickering emulsions (NSSPEs), (**B**) appearance of reconstituted emulsions from solid NSSPE prepared with different solid carriers after standing for 8 h at room temperature and (**C**) droplet size of the liquid NSSPE and reconstituted emulsions (RE) from solid NSSPEs (mean ± SD, *n* = 3).

**Figure 3 molecules-26-01809-f003:**
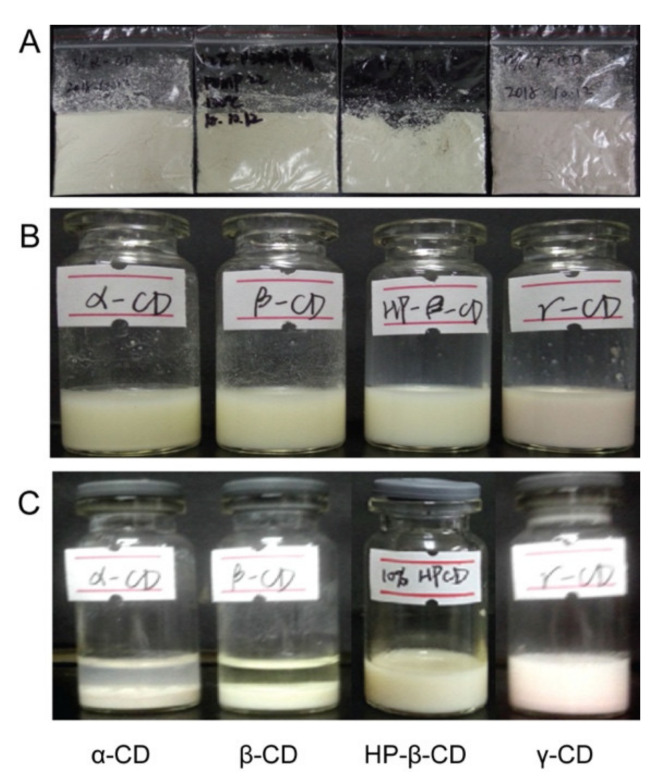
Appearance of (**A**) the solid NSSPEs and (**B**) liquid NSSPEs with different type of cyclodextrin and (**C**) corresponding reconstituted emulsions of solid NSSPEs.

**Figure 4 molecules-26-01809-f004:**
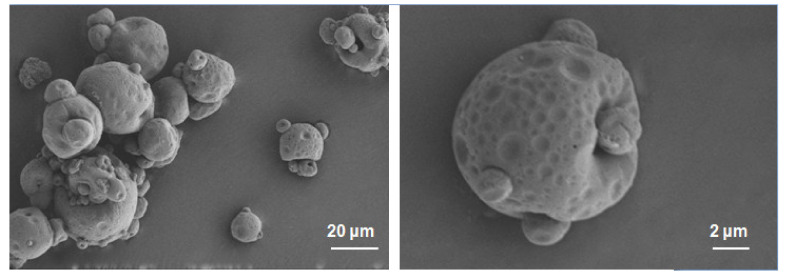
SEM images of spray-dried puerarin nanocrystals self-stabilized Pickering emulsion.

**Figure 5 molecules-26-01809-f005:**
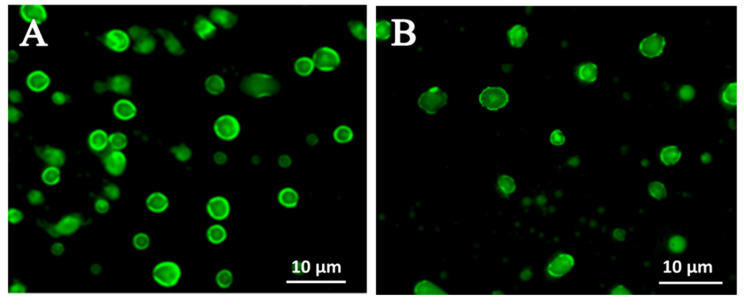
Fluorescence micrographics of (**A**) the liquid NSSPE and (**B**) reconstituted emulsion from the solid NSSPE.

**Figure 6 molecules-26-01809-f006:**
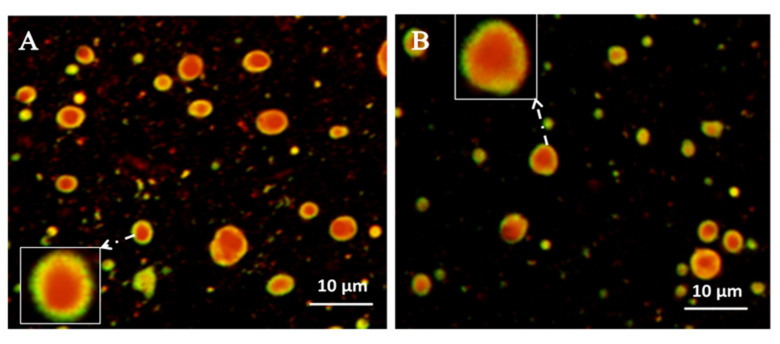
Confocal laser scanning microscopy images of(**A**) the liquid NSSPE and (**B**) reconstituted emulsion from the solid NSSPE.

**Figure 7 molecules-26-01809-f007:**
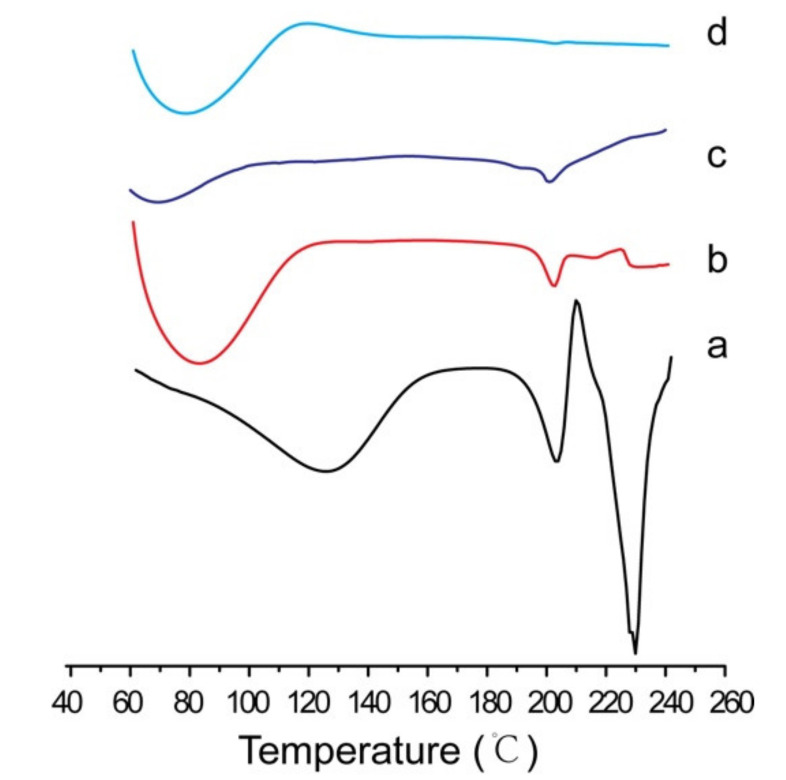
Differential scanning calorimetry thermogram of (**a**) puerarin crude material, (**b**) puerarin nanocrystals, (**c**) adsorbed puerarin in liquid NSSPE and (**d**) adsorbed puerarin in reconstituted emulsion from the solid NSSPE with HP-β-CD as carrier.

**Figure 8 molecules-26-01809-f008:**
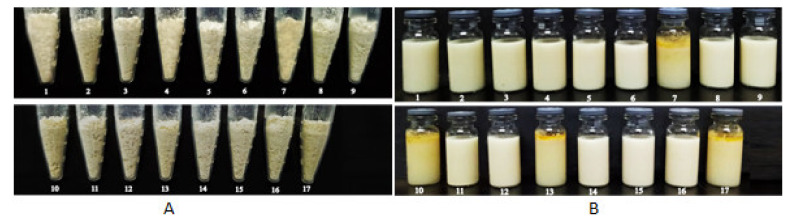
Appearance of (**A**) the solid NSSPE and (**B**) their corresponding reconstituted emulsions prepared as Box–Behnken design.

**Figure 9 molecules-26-01809-f009:**
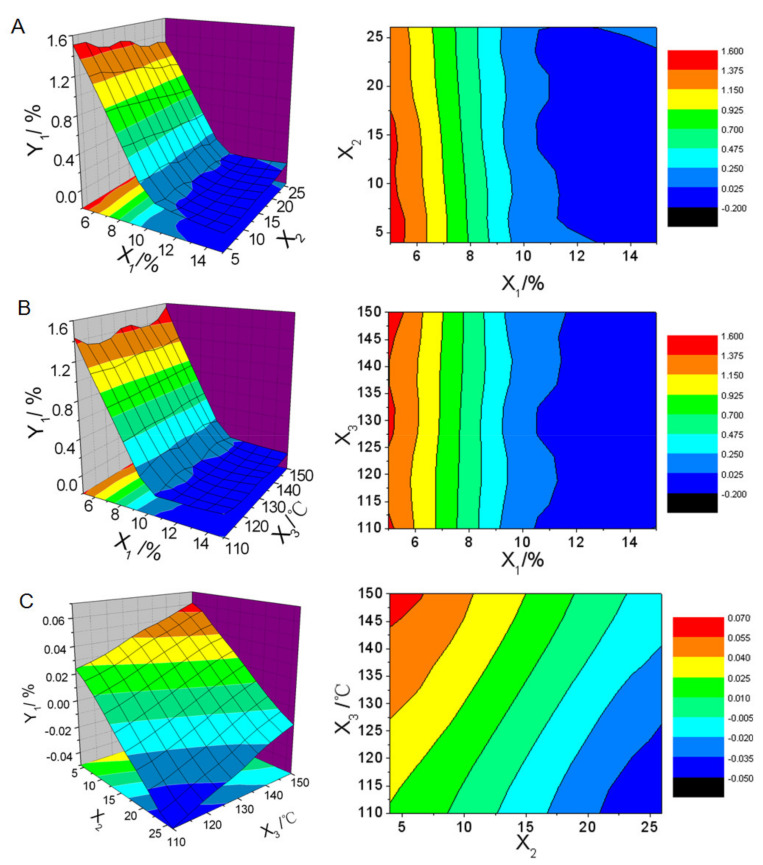
Three-dimensional response surface plots and contour maps of oil leakage rate of reconstituted emulsion (*Y*_1_) in function of (**A**) amount of solid carrier (*X*_1_) and feed rate (*X*_2_), (**B**) *X*_1_ and inlet air temperature (*X*_3_)), and (**C**) *X*_2_ and *X*_3_.

**Figure 10 molecules-26-01809-f010:**
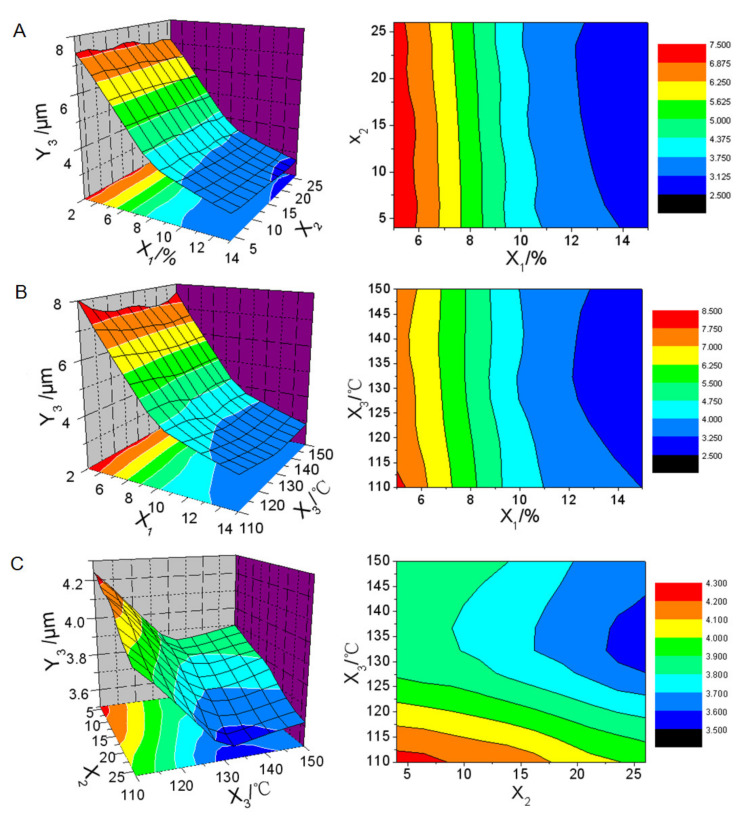
Three-dimensional response surface plots and contour maps of droplet size of reconstituted emulsion (*Y*_2_) in function of(**A**) amount of solid carrier (*X*_1_) and feed rate (*X*_2_), (**B**) *X*_1_ and inlet air temperature (*X*_3_), and (**C**) *X*_2_ and *X*_3_.

**Figure 11 molecules-26-01809-f011:**
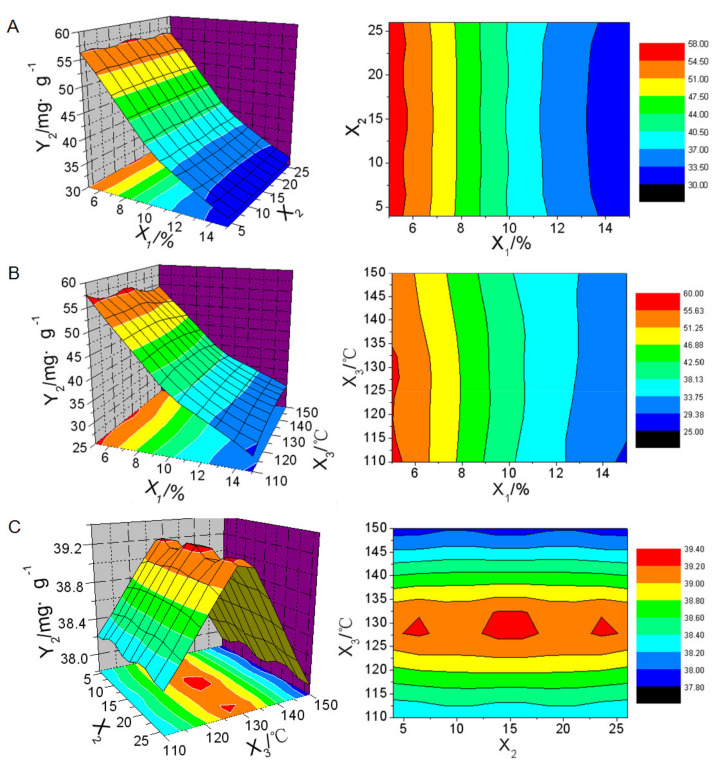
Three-dimensional response surface plots and contour maps of puerarin content in solid NSSPE (*Y*_3_) in function of (**A**) amount of solid carrier (*X*_1_) and feed rate (*X*_2_), (**B**) *X*_1_ and inlet air temperature (*X*_3_), and (**C**) *X*_2_ and *X*_3_.

**Figure 12 molecules-26-01809-f012:**
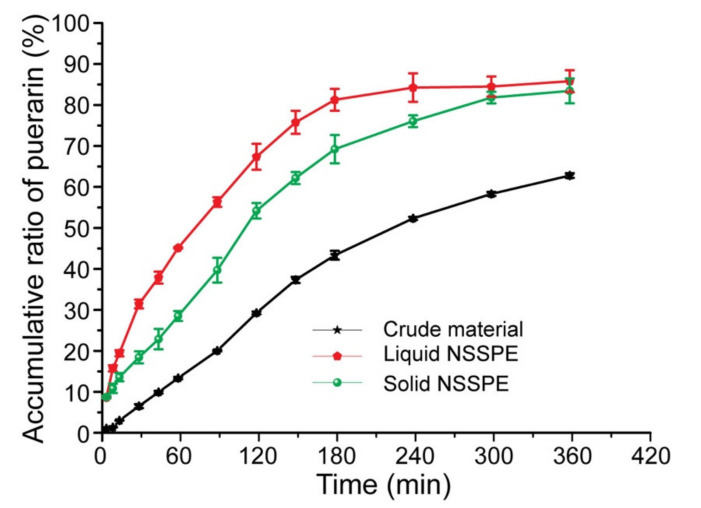
The dissolution profiles of puerarin crude material, the liquid NSSPE and the solid NSSPE. Data are expressed as mean ± SD (*n* = 3).

**Figure 13 molecules-26-01809-f013:**
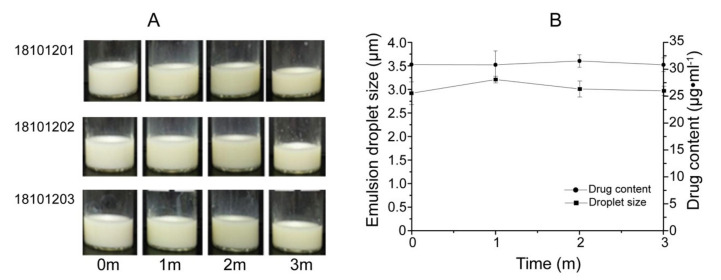
The appearance (**A**), droplet size and drug content (**B**) of reconstituted emulsion from the solid NSSPE during three-month storage at 40 °C, 75% of relative humidity (*n* = 3).

**Table 1 molecules-26-01809-t001:** Comparison of the liquid NSSPE and reconstituted emulsion from the solid NSSPE (mean ± SD, *n* = 3).

Sample	Emulsion Droplet Size/μm	Particle Size of Adsorbed Nanocrystals/nm	Adsorption Rate of Nanocrystals/%
the liquid NSSPE	3.05 ± 0.38	264.4 ± 9.8	29.3 ± 0.5
Reconstituted emulsion from the solid NSSPE	2.81 ± 0.14	280.9 ± 10.9	26.8 ± 1.7

**Table 2 molecules-26-01809-t002:** The experimental arrangement and results of Box-Behnken design.

Number	Amount of Solid Carrier (*X_1_*)	Feed Pump Parameter (*X_2_*)	Inlet Air Temperature (*X_3_*)/°C	Oil Leakage Rate (*Y_1_*)/%	Droplet Size of Reconstituted Emulsion (*Y_2_*)/μm	Drug Content in Solid NSSPE (*Y_3_*)/mgg^−1^
1	0(10%)	0 (15)	0(130)	0	4.08	38.95
2	0(10%)	0(15)	0(130)	0	3.65	38.68
3	0(10%)	0(15)	0(130)	0	3.57	38.65
4	0(10%)	0(15)	0(130)	0	4.00	38.79
5	0(10%)	0(15)	0(130)	0	4.49	38.86
6	0(10%)	1(25)	−1(110)	0	4.04	38.56
7	−1(5%)	0(15)	1(150)	1.52	8.04	53.05
8	0(10%)	1(25)	1(150)	0	3.55	37.78
9	0(10%)	−1(5)	1(150)	0	3.61	39.08
10	−1(5%)	1(25)	0(130)	1.39	7.16	60.07
11	1(15%)	0(15)	−1(110)	0	3.00	29.56
12	1(15%)	1(25)	0(130)	0	2.62	29.90
13	−1(5%)	−1(5)	0(130)	1.69	7.31	57.53
14	1(15%)	0(15)	1(150)	0	2.78	31.59
15	1(15%)	−1(5)	0(130)	0	3.10	30.53
16	0(10%)	−1(5)	−1(110)	0	4.28	37.87
17	−1(5%)	0(15)	−1(110)	1.36	8.18	56.48

**Table 3 molecules-26-01809-t003:** Predicted and observed values of responses based on the optimized levels of independent variables(*X*_1_ = 12%, *X*_2_ =23, *X*_3_ =130, *n* = 3).

Response	Oil Separation Rate (*Y_1_*)/%	Droplet Size (*Y_2_*)/μm	Drug Content in the Solid NSSPE (*Y_3_*)/mg·g^−1^
Predictive value	−01779	2.85	34.91
Measured value	0	2.81 ± 0.14	31.26 ± 0.63

## Data Availability

Data available in a publicly accessible repository.

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
