# Peer review of "A Novel Solid Nanocrystals Self-Stabilized Pickering Emulsion Prepared by Spray-Drying with Hydroxypropyl-β-cyclodextrin as Carriers"

_molecules, 2021, doi:10.3390/molecules26061809_

Round 1
Reviewer 1 Report
Dear Editor,
The contribution by J. Zhang and coworkers presents experimental findings aiming at developing a new solid drug nanocrystals self-stabilized Pickering emulsion (NSSPE) of puerarin using spray drying in combination with water soluble solid carriers. This work is a continuation of a previous report (ref. 2) in which a liquid NSSPE of puerarin was prepared and proved to improve oral bioavailability of this molecule. Now, authors have attempted to find an appropriate solid carrier able to retain the special features of liquid NSSPE. They succeeded in this goal by obtaining a new solid form of NSSPE of puerarin with HP-β-CD as carrier. As part of the characterization of this material, fluorescence microscopy, confocal laser scanning microscopy and scanning electron microscopy was used.
I think this is an interesting piece of research and I find it suitable for publication in Molecules with only one observation that I detail below:
The “selection process” to choose the solid carrier was an empirical “trial-and-error” essay guided by the appearance of the emulsion (Figs. 1 and 2). In the section 3. Results and discussion (p.6 and 7), authors have attempted to give some reasons why HP-β-CD worked out as carrier whereas other CDs did not. However, these ideas were quite general and the “inclusion effect” invoked also apply to the other CDs. From a reader standpoint, out of this discussion it was not possible to learn why HP-β-CD worked out better. Please elaborate a further explanation to understand the molecular basis of the favorable effect of HP-β-CD in contrast to the other CDs employed.
As an added comment, knowing the structural features of puerarin and HP-β-CD it would be of help. Otherwise, it will be curious for a journal named Molecules not to portray the molecular structures of puerarin and HP-β-CD.
Author Response
Dear reviewer,
Thank you for providing us the valuable comments and suggestions on our manuscript. We value your comments and suggestions and address each of the points in sequence below. All the modified words or sentences have been incorporated in the revised manuscript and these modifications have been marked in red.
Best regards
Jifen Zhang

Reviewer 2 Report
To the authors,
The current manuscript is aimed to use Pickering emulsion preparation method to increase the bioavailability of puerarin. The technique is the spray drying method used in this study seems to be interesting for the readers and a practical approach to address the main question of this study, however, the authors should validate the crystallinity of the API after formulation with a pH of 11.5. At least an XRD analysis with a proper control could validate the claim of the authors. I believe that this data should be added. All in all, the methodology is quite well structured. The discussion should also talk further about previous studies and compare the results with other data published before.
Regards
Author Response

(The authors gave the same response as above.)
